# Interaction of Soybean Varieties and Heat Treatments and Its Effect on Growth Performance and Nutrient Digestibility in Broiler Chickens

**DOI:** 10.3390/ani11092668

**Published:** 2021-09-10

**Authors:** Florian Hemetsberger, Thomas Hauser, Konrad J. Domig, Wolfgang Kneifel, Karl Schedle

**Affiliations:** 1Department of Food Science and Technology, Institute of Food Science, University of Natural Resources and Life Sciences Vienna, 1190 Vienna, Austria; florian.hemetsberger@boku.ac.at (F.H.); konrad.domig@boku.ac.at (K.J.D.); wolfgang.kneifel@boku.ac.at (W.K.); 2Department of Agrobiotechnology, Institute of Animal Nutrition, Livestock Products and Nutrition Physiology, University of Natural Resources and Life Sciences Vienna, 1190 Vienna, Austria; thomas.hauser@students.boku.ac.at

**Keywords:** soybean, broiler, heat treatment, trypsin inhibitor, protein solubility, amino acid digestibility

## Abstract

**Simple Summary:**

Soybeans are the major source of protein in today´s livestock diets. However, European soybean imports are under criticism because of environmental issues. Therefore, production of European soybeans is expected to grow. To ensure optimal feeding properties, soybeans require a heating process to eliminate intrinsic compounds interfering negatively with the animal’s digestive tract. A heating process might have different effects on different soybean varieties. Therefore, two different soybean varieties were treated at two different heat intensities in the present study—110 °C and 120 °C. The results showed, that while both heat intensities had a sufficient deactivating effect in one variety, the other variety was not treated sufficiently at 110 °C. Insufficient heat treatment was expressed in lower weight gains and lower feed intake. No negative effect of heat treatment at 120 °C was observed for growth performance, but amino acid digestibility was reduced. The present study shows that the optimal processing conditions can vary for different soybean varieties, which has to be considered especially when handling small and heterogeneous soybean batches.

**Abstract:**

As production of European soybeans is expected to grow, optimal processing conditions need to be ensured for small and heterogeneous batches of soybeans. The effect of different soybean varieties, as well as heat treatments, on the growth performance and nutrient digestibility in broiler chickens was investigated. Two varieties, regarded as heat stable and heat labile after preliminary experiments, were partially de-oiled and thermally processed at 110 °C for 20 min and 120 °C for 20 min. The resulting soybean cakes were integrated into a mash diet and subjected to a 36-day long feeding experiment. A total of 336 one-day-old broiler chickens were divided into 24 pens, resulting in 6 replicates per treatment. With application of the 110 °C treatment, analysis of soybean cakes showed that the commonly required reduction in trypsin inhibitor activity (TIA) was only reached with one soybean variety. The higher processing temperature of 120 °C ensured sufficient TIA reductions in both soybean varieties. Elevated TIA concentrations resulted in decreased growth performances (*p* < 0.05) of the chickens, whereas no negative effect from overheating on growth performance appeared. Total-tract nitrogen retention (*p* < 0.05) and pre-caecal digestibility of several amino acids (*p* < 0.10) decreased with higher processing temperatures but had no negative effects on growth performance. In conclusion, the results indicate that processing conditions adjusted to the different varieties are essential to ensure optimal product quality.

## 1. Introduction

Feedstuffs based on soybeans are an essential part in the diets of livestock animals, especially for monogastrics such as pigs and poultry. Worldwide demand, however, is mainly supplied by large-scale producers in North and South America, which are associated with deforestation and attract criticism for additional reasons, such as excessive use of pesticides [1]. To increase their autonomy on protein feeds and counteract environmental issues associated with imported soybeans, European countries have promoted an increase in local production of soybeans and other protein-delivering plants [2]. In Austria, soybeans have become a well-established crop, ranking behind only maize, wheat and barley [3]. In the European Union, cultivation of soybeans constitutes 2% of arable land, although this share is expected to increase [4]. Besides expanding cultivation of soy, an optimized processing infrastructure should be established, as large, optimized processing plants are currently located far from all growing regions. Local small-scale processing might therefore possess the potential to accelerate the protein self-sufficiency of farmers. It is well known that optimum processing of soybeans is crucial for providing sufficient feeding quality. In native soybeans, antinutritive factors such as trypsin inhibitors are of crucial relevance, as they exert adverse effects on the digestive process in the gastrointestinal tract. The negative effects of the trypsin inhibitor on monogastric animals are well described in literature [5,6]. Hence, such effects need to be minimized by tailored heat treatment [7]. Several experiments engaged in the task of finding processing condition ensuring optimal feeding suitability [8,9,10,11]. According to the studies undertaken, trypsin inhibitor activity (TIA) of soybeans must be reduced to 3–4 g/kg to ensure optimal animal growth. However, findings of individual studies revealed quantities other than the recommended TIA of 3–4 g/kg to be acceptable—lower or higher—emphasizing the fact that properties other than trypsin inhibitors also contribute to soybean´s feeding suitability [5,12,13]. A plurality of methods is used to minimize TIA, such as extrusion [14], roasting over direct fire [15], heated air or by infrared radiation [16]. However, most soybeans undergo heat treatment after a solvent extraction step to separate the soybean oil [17]. In the course of solvent recovery, direct and indirect steam heats the extracted commodity to 100–110 °C [18]. Although heat is an essential step in soybean processing, excessive heat treatment can result in decreased nutrient digestibility, especially for protein due to the formation of other potentially harmful substances such as Maillard products [6]. Owing to these factors and experiences, local and decentral processing plants have been challenged by the high variability of nutrients in raw soybeans, as they may in turn affect optimum processing conditions needed to ensure a highly digestible feed compound [19,20]. Besides processing temperature and time, factors such as moisture, bean or particle size, dehulling and the presence of reducing substances interact with the soybean´s response to heat [13,21,22,23,24]. Useful indicators have thus been sought to assess the influence of thermal processing on the protein quality of soybean products. Among these, protein solubility in KOH has been identified as a simple and inexpensive method to estimate protein degradation caused by overprocessing [25]. Nevertheless, other methods such as direct analysis of the amino acid profile, reactive lysine and Maillard products were also implemented in analytical monitoring [26]. 

Considering the critical issues mentioned above, the goal of this study was to investigate the effect of two selected soybean varieties possessing different sugar levels and KOH solubility and treated at different heat intensities on the performance and nutrient digestibility of broiler chickens.

## 2. Materials and Methods

### 2.1. Animal Housing and Feeding

As usual under practical conditions, 336 Ross 308-day-old mixed-sex chickens were obtained from a commercial hatchery. They were weighed and divided into 24 pens of 14 animals each, resulting in six replicates per treatment. Pens of 3 m^2^ were equipped with flat chain feeders and bell drinkers to provide ad libitum access to feed and water; wood shavings were applied as litter. Feed was provided in a ground form using a hammermill of 4.5 mm in sieve diameter. Temperature was regulated continuously from 30 °C on day 1 to 20 °C on day 36. A lighting program increased dark phases from 0 to 6 h from day 1 to 7 and kept constant afterward. On day 15, all birds weighing less than 75% of the pen’s average were removed from the trial. The feeding trial was divided into starter (days 1–14), grower (days 15–28) and finisher phases (days 29–36). At the end of the experiment, animals were stunned per percussive blow to the head and bled immediately afterward. The experiment was conducted in compliance with the First Ordinance of Animal Husbandry of the Austrian Federal Ministry of Health and Women (BGBI. II Nr. 485/2004). The experiment was reviewed and approved by the Ethics Committee of the University of Natural Resources and Life Sciences, Vienna (reference number 2021/004).

### 2.2. Diets

In preliminary tests, seven soybean varieties were screened for their response to heat treatment. Beans were ground through a 0.5 mm sieve and heated for 60 min at 105 °C in a laboratory autoclave (Autoclave/steam sterilizer 2540ELV, Tuttnauer, Breda, Netherlands), in addition to heating and cooling phases of 20 and 10 min, respectively. Heat treatments were performed in duplicates for each variety, and obtained samples were pooled. Soybeans were then analyzed for their protein solubility in a 2 g/L KOH solution. Of the soybeans tested, the two varieties showing the greatest disparity were selected. Variety 1 lost about 4% of its initial protein solubility of 93% and was therefore considered and denominated as the heat-stable variety (SV). Variety 2, also initially 93%, lost 19% and was considered the heat-labile variety (LV). For the feeding trial, partially de-oiled soybeans of these varieties were treated in an autoclave at 110 °C for 20 minutes, plus 20 minutes for heating up and cooling down, to reduce TIA to a value of 3–4 mg/kg. To simulate heat damage, soybean cakes were heated for 20 minutes at 120 °C, plus 45 minutes of heating and cooling. All diets were calculated to meet the recommended requirements of the breeder [27] and consisted of a basal mixture containing corn and soybean meal as its main components, supplemented with 300 g/kg of experimental soybean cake (Table 1). The experiment was conducted as a 2 × 2 factorial study concerning the factors soybean variety and heat treatment. Six replicate pens were available for each experimental diet.

### 2.3. Performance Parameters

After each period, birds from each pen were weighed, and feed consumption was determined. Mortalities were recorded with date and weight. Average daily gain (ADG) was calculated pen-wise after each feeding phase by dividing weight gain per pen (inclusive of weight of mortalities) by the fattening days of all animals per pen (inclusive of mortalities). Average daily feed intake (ADFI) was calculated by dividing consumed feed per pen by the fattening days of all animals per pen (inclusive of mortalities). Feed conversion rate (FCR) was estimated by dividing ADFI by ADG. On the day of slaughtering, weights of the eviscerated carcass (excluding blood, feathers, giblets and intestinal tract), abdominal fat, heart, liver and gizzard of every bird were recorded. Cooled carcasses with and without head/neck were weighed again. Two representative animals, one female and one male, from each pen were selected to collect the weights of the breast, head and neck, feet, thigh and drumstick as well as wings.

### 2.4. Sample Collection and Chemical Analysis

After slaughtering, the pancreas with its adherent section of the duodenum of six representative animals, three female and three male, was collected and stored in formalin. After one week, the pancreas was separated from the gut and weighed. These six animals were also used to collect digesta of the proximal 2/3 of ileum for amino acid digestibility analysis. Digesta was pooled in boxes and stored in dry ice immediately after collection. On day 33, excreta of all birds were collected for determination of total-tract apparent digestibility. For this purpose, excreta were frozen immediately at −20 °C and freeze-dried before analysis. Samples of soybean cakes, experimental diets and dried excreta were ground to pass a 1 mm screen and analyzed according to standard procedures for the parameters dry matter, ash, crude protein, crude fiber, ether extract, ether extract after acid hydrolysis, sugar, starch, phosphorous, sodium, potassium and calcium [29]. Amino acid content of ileal digesta and feed was analyzed by applying the methods of Altmann (1992), using high-performance liquid chromatography after a 20 h hydroxylation process with 6 M HCl and a previous stabilization by use of tryptophan with Ba(OH)_2_ [30]. Separation of amino acids was accomplished using a hyperphil octadecylsilyl 250 × 4 mm column after pre-column derivatization with orthophtalaldehyde. TiO_2_ as an inert marker was analyzed according to Jagger et al. (1992) [31]. Protein solubility in KOH of the experimental soybean cakes was determined according to Araba and Dale (1990) [25]. TIA was measured following the procedure by Austrian Standard [32]. Gross energy content of samples was analyzed using bomb calorimetry (IKA C200, Staufen, Germany). Apparent metabolizable energy (AMEn) of diets was calculated according to the formula of GfE (1999) [28].

The coefficient of apparent total-tract apparent digestibility (ATTD) of nutrients, dry matter and gross energy as well as pre-caecal digestibility was determined using their respective ratios to TiO_2_ in feed and excreta/digesta and applying the following formula: 1 – (TiO_2_ in feed*nutrient in excreta or digesta)/(TiO_2_ in excreta or digesta*nutrient in feed).

### 2.5. Statistical Analysis

Data were analyzed using the MIXED procedure of SAS 9.4 (SAS Institute Inc., Cary, NC, USA). Chicken pens were the experimental units for performance parameters and digestibility, as well as animals for the carcass traits. The model used variety, heat treatment and their interaction as fixed effects for the performance and digestibility parameters. For carcass characteristics, the sex of the animals was additionally included in the model. Mean values of groups were displayed as least square means. Significant differences were considered for *p* < 0.05 and statistical trends at *p* < 0.10. Tukey’s test was used for post hoc analysis.

## 3. Results

### 3.1. Soybean Cake

Compositional details of the soybean cakes are listed in Table 2. Soybean cakes had similar dry matter irrespective of heat treatment or variety. The crude protein content of SV cakes was 40 g/kg higher than that of the LV, while the sugar content was about 20% lower. Low heat-treated soybean cakes showed comparable protein solubility values in KOH. High heat treatment induced a more distinctive decline in the LV than in SV. Following low heat treatment, the LV had a TIA above the recommended level of 3–4 g/kg [14,18], whereas the low heat-treated SV was within the range. High heat treatment resulted in a TIA below the detection limit. In diets with the LV, the energy content of the compound feed was about 0.2 MJ AMEn lower than with the SV. Neither soybean variety nor heat treatment had an effect on the protein content of the diets of each phase. Diets using the high heat-treated cake had a TIA below 1.0 g/kg. The TIA of diets with low heat-treated cakes was between 1.5 and 1.6 for the SV and between 1.8 and 2.1 mg/g for the LV, respectively.

### 3.2. Animal Performance

In the starter phase, animals fed diets containing SV cakes measured about 4% higher average daily gain (ADG) and average daily feed intake (ADFI) (*p* < 0.05) (Table 3). In the grower phase, diets with the SV resulted in about 5% higher ADG and ADFI (*p* < 0.05). This effect disappeared in the finisher phase. Regarding the entire fattening period, however, chickens receiving the SV revealed a higher ADG and ADFI (*p* < 0.05). Confirmed by a statistically significant interaction between heat treatment and variety, ADFI and ADG decreased in the LV low-temperature treatment compared with all others over the fattening period (*p* < 0.05). In the finisher phase, high heat treatment increased ADG and ADFI (*p* < 0.05), which, over the entire feeding period, in addition to body weight at day 36, were highest with the SV cakes and high-temperature treatment. A significant interaction between variety and temperature was therefore demonstrated regarding ADG and ADFI (*p* < 0.05). Lastly, for groups receiving high heat-treated soybean cakes, mortalities were significantly higher over the entire feeding period (*p* < 0.05).

### 3.3. Carcass Characteristics

Relative yield of eviscerated carcass (EC), cold carcass (CC) and carcass for grilling (carcass without feet, head and neck) in terms of live weight showed no differences between treatments (Table 4). Groups fed with the SV generated 28% more abdominal fat, and those with the LV had 9% higher relative liver weights (*p* < 0.05). Relative breast weights were 9% higher when animals were fed with low heat-treated soybeans (*p* < 0.05). Significant differences in carcass characteristics were also observed for heart, feet and wing weights (*p* < 0.05). The highest pancreas weight was observed for LV diets treated at 110 °C and the lowest with the SV treated at 120 °C, offering evidence of significant interaction between variety and heat treatment (*p* < 0.05). Regarding sex, significant differences between male and female animals were observed for the parameters body weight (BW), EC, CC, carcass for grilling, as well as abdominal fat-, heart- and pancreas percentage (*p* < 0.05; Table 4).

A significant (*p* < 0.05) interaction between variety, heat treatment and sex appeared for the parameter EC. Similar results were observed for the male and female birds. The heat-labile low treatment showed significant lower weights compared with the other treatments. A similar trend was observed for the parameter BW, CC and abdominal fat percentage.

### 3.4. Digestibility

Nitrogen retention was significantly higher with low heat-treated soybeans (*p* < 0.05) (Table 5). The coefficient of ATTD for starch was affected by variety and heat treatment (*p* < 0.05), as SV soybeans treated at 110 °C notably had the highest starch digestibility. No differences in ATTD occurred for dry matter, ash, ether extract or gross energy.

Heat treatment influenced ileal apparent digestibility of crude protein and the amino acids lysine, histidine, proline and glutamic acid according to the data of Table 6 (*p* < 0.10). Digestibility of aspartic acid and cysteine was affected significantly (*p* < 0.05) by heat treatment. Ileal analysis showed no effects for other amino acids, dry matter or ammonia and no effects of soybean variety.

## 4. Discussion

Regarding protein solubility, all soybean cakes were under the upper threshold of 85% recommended by Araba and Dale (1990) [33], but sufficient deactivation of trypsin inhibitors was not achieved in all samples. This confirms the assumption that protein solubility in KOH should not be used as a single parameter to determine whether heat treatment is sufficient or not. The protein solubility of high heat-treated soybean cakes was lower than 70%, classifying them therefore as overprocessed [25]. Nevertheless, it has to be taken into account that the measurement of protein solubility is highly impacted by particle size [34]. Consequently, a standardized sample preparation is essential to make protein solubility a robust parameter, which is of particular interest, given that different grinding screens are used in literature [25,35]. Analytical data of soybean cakes showed that even though the decline of protein solubility was more distinctive for the LV cakes, TIA was higher when treated at a lower heat intensity. This agrees with several studies that demonstrated soybeans of different origin or genetics requiring different treatments to ensure optimal feed quality [20,36]. The response of soybean cake in this experiment was not caused by particle size and moisture, as raw soybeans had identical moisture contents and were processed equally. Hence, the different responses of the soybeans might be attributed to intrinsic factors such as reducing substances, e.g., reducing sugars [24].

Broilers fed with a diet including the low-temperature-treated LV had significantly lower feed intake and weight gains than those involving all other treatments. This indicates that the TIA of 5.1 g/kg in soybean cakes and approximately 2 g/kg in the diet, respectively, exerted a negative effect on the growth performance of the animals. This finding has also been supported by several studies and is in agreement with the recommended TIA of 3–4 g/kg [5,8,9,10,11]. Tolerances for higher TIA, as demonstrated in other studies, could not be confirmed [12,13]. The hypothesis that TIA is the most relevant factor causing growth depression can be supported by increased pancreas weight and has been proposed by other researchers [5,14]. Nevertheless, in our study, lower weight gains were mainly caused by lower feed intake. No difference in feed conversion rate appeared. These findings are in contrast to other studies where negative effects on FCR were observed [5,37]. Other authors even saw maximum feed intake when raw soybeans were fed [11]. In agreement with our results, Woyengo et al. (2017) [38] reported reductions in feed intake with diets high in trypsin inhibitors. They hypothesized that the increased cholecystokinin production needed to stimulate trypsin secretion causes a reduction in voluntary feed intake. Reduced amino acid digestibility has also been suggested as a cause of decreased feed intake [38]. However, this hypothesis contrasts with our findings, as no negative effect of TIA on pre-caecal amino acid digestibility was found. The higher percentage of abdominal fat for groups that were fed SV cakes corresponds with the higher AMEn content in these diets [39].

Lower nitrogen retention for high-temperature-treated soybean cakes indicates lower protein digestibility for these treatments, although no negative effect on growth performance was observed. These findings might indicate sufficient supply of essential amino acids in all experimental diets. On the other hand, in the case of a suboptimal supply with essential amino acids, negative effects on performance would have been expected [40]. Findings involving pre-caecal digestibility may relate to some heat damage of the heat-labile amino acids cysteine, lysine, proline, histidine, glutamic acid and asparagine as well as crude protein in soybeans treated with the higher temperature. This interpretation is also backed by the fact that most of the amino acids affected are prone to heat damage because their side chains possess a thiol or amino group [26]. In agreement with our findings, Parsons et al. identified cysteine, lysine and asparagine as amino acids most prone to heat damage [41]. Nevertheless, no interaction was detected for the pre-caecal apparent digestibility of the analyzed amino acids, indicating a more pronounced effect of the heat treatment [40]. Hence, it must be considered that elevated pancreatic secretions and endogenous amino acid losses caused by trypsin inhibitors could interfere with amino acid digestibility analysis [42]. 

One reason for the observed, although numerically small, differences in starch digestibility may be the fact that amylose–lipid complexes were generated in the high-temperature-treated varieties [43]. Higher mortality in the high heat treatment groups could be linked to higher concentrations of Maillard reaction products in these treatments. The negative effects of these products on health parameters have been frequently discussed [44]. However, no effects on poultry have been identified so far.

The well-described sexual dimorphism between male and female broiler chickens of increased ADG and ADFI was also recorded in the present study [45].

## 5. Conclusions

Complying with the recommendations, a TIA under 3–4 g/kg is crucial for ensuring high animal-growth performance. Despite lower protein digestibility caused by higher heat treatment without an observed interaction of the variety, the TIA of the diet can be regarded as the decisive factor for high growth performance. Precondition is a sufficient supply with essential amino acids according to the recommendation of the breeder. On the other hand, protein solubility as the sole parameter to estimate sufficient heat treatment failed to act as an indicative value for optimal growth [34].

The present study shows that sufficient heat treatment should be tailored and thus differs for diverse soybean varieties. Furthermore, this might also be valid for different growing conditions. Optimum processing parameters must be monitored for different soybean batches if sufficient product quality is to be guaranteed.

## Figures and Tables

**Table 1 animals-11-02668-t001:** Compositional details of the diets applied.

Phase	Starter				Grower				Finisher			
Variety	Heat-Stable		Heat-Labile		Heat-Stable		Heat-Labile		Heat-Stable		Heat-Labile	
Heat Treatment	Low	High	Low	High	Low	High	Low	High	Low	High	Low	High
Ingredients, g/kg.												
Corn	486.6		445.6		502.0		461.0		532.3		491.3	
Soybean meal HP	129.0		165.3		111.3		147.5		63.8		100.1	
SV low T	300.0				300.0				300.0			
SV high T		300.0				300.0				300.0		
LV low T			300.0				300.0				300.0	
LV high T				300.0				300.0				300.0
Sunflower oil	17.0		23.2		27.1		33.3		39.8		46.0	
Lignocellulose	10.0		10.0		10.0		10.0		10.0		10.0	
Calcium carbonate	12.3		13.5		11.7		12.8		15.5		16.6	
Dicalcium phosphate	13.6		11.4		11.1		8.9		9.4		7.3	
Sodium chloride	3.5		3.4		3.5		3.5		3.5		3.5	
L-Lysine HCl	5.0		4.8		3.8		3.6		3.6		3.3	
DL-Methionine	4.6		4.7		3.9		4.0		3.6		3.6	
L-Threonine	2.6		2.5		1.9		1.8		1.7		1.6	
L-Arginine	1.5		1.5		0.5		0.5		0.4		0.3	
L-Isoleucine	1.2		1.1		0.6		0.6		0.6		0.6	
L-Valin	1.8		1.7		1.1		1.1		1.0		0.9	
Mineral/vitamin premix ^a^	10.0		10.0		10.0		10.0		10.0		10.0	
Choline-Cl	0.8		0.8		0.8		0.8		0.8		0.8	
Phytase ^b^	0.1		0.1		0.1		0.1		0.1		0.1	
Coccidiostat ^c^	0.5		0.5		0.5		0.5		-		-	
Titanium dioxide	-		-		-		-		4.0		4.0	
Calculated composition, g/kg												
AMEn, MJ/kg	12.6		12.6		13.0		13.0		13.4		13.4	
Crude protein	230.5		230.5		220.0		220.0		200.0		200.0	
Dig. Lys	12.8		12.8		11.5		11.5		10.3		10.3	
Dig. Met	7.0		7.0		6.2		6.3		5.7		5.8	
Dig. Thr	8.6		8.6		7.7		7.7		6.9		6.9	
Dig. Trp	2.0		2.0		1.9		1.9		1.7		1.7	
Dig. Arg	13.7		13.7		12.3		12.3		11.0		11.0	
Analyzed composition, g/kg												
Dry matter	892	892	895	896	891	890	896	895	894	894	899	899
Crude protein	239	233	238	237	217	225	221	228	205	206	203	208
Starch	352	357	328	332	352	366	344	339	383	379	356	347
Ether extract	75	76	78	80	84	82	90	88	99	98	102	103
Sugar	41	40	51	46	43	40	47	46	36	36	45	46
Crude fiber	28	31	35	44	27	28	34	41	29	39	35	33
Ash	66	66	67	67	63	60	64	63	63	67	69	66
Calcium	11	13	12	13	11	11	11	10	13	12	13	12
Phosphor	6.4	6.7	6.7	6.8	6.0	6.3	5.4	5.9	5.7	5.9	5.8	5.8
Sodium	1.5	1.4	1.4	1.7	1.5	1.4	1.4	1.2	1.5	1.4	1.6	1.4
Potassium	9.7	9.9	10.0	10.5	8.7	8.9	9.3	9.6	8.3	8.3	9.1	9.4
AMEn, MJ/kg ^d^	12.7	12.7	12.5	12.5	12.7	13.0	12.9	12.8	13.4	13.3	13.2	13.1
TIA	1.5	0.8	2.1	0.8	1.5	0.8	2.1	0.9	1.6	0.6	1.8	0.8
Lys	n.a.	n.a.	n.a.	n.a.	n.a.	n.a.	n.a.	n.a.	13.1	13.1	13.6	16.6
Met	n.a.	n.a.	n.a.	n.a.	n.a.	n.a.	n.a.	n.a.	6.2	6.1	6.4	6.4
Thr	n.a.	n.a.	n.a.	n.a.	n.a.	n.a.	n.a.	n.a.	8.7	8.8	9.0	9.3
Arg	n.a.	n.a.	n.a.	n.a.	n.a.	n.a.	n.a.	n.a.	13.5	13.5	13.3	13.6

Abbreviations: SV, heat-stable variety; LV, heat-labile variety; AMEn, apparent metabolizable energy; TIA, trypsin inhibitor activity; n.a., not analyzed. ^a^ Composition per kg premix: 1,000,000 IU vitamin A, 400,000 IU vitamin D3, 2000 mg vitamin E, 400 mg vitamin K, 300 mg vitamin B1, 750 mg vitamin B2, 450 mg vitamin B6, 2250 µg vitamin B12, 6900 mg nicotinic acid, 1950 mg pantothenic acid, 195,000 µg folic acid and 12,000 µg biotin; 1680 mg Fe, 8000 mg Zn, 10,000 mg Mn, 1200 mg Cu, 100 mg I, 25 mg Se; ^b^ Optiphos^®^ 2500 CT 6-phytase derived from *E. coli* (Huvepharma, Sofia, Bulgaria), 250 OTU per kg diet; ^c^ Sacox 120^®^ micro Granulate (Huvepharma, Sofia, Bulgaria); ^d^ calculated according to the formula for compound feed of the society of nutrition physiology [28].

**Table 2 animals-11-02668-t002:** Analytical data of soybean cakes.

Variety	Heat-Stable		Heat-Labile	
Heat Treatment	Low	High	Low	High
Analyzed values, g/kg				
Dry matter	886	885	894	905
Ash	54.8	52	58.8	58
Crude protein	402	400	358	363
Ether extract	130	132	124	128
Crude fiber	46	48	53	50
Sugar	78	75	95	93
Starch	65	66	67	66
Phosphor	71	65	69	56
Calcium	14	14	22	21
Sodium	n.d.	n.d.	n.d.	n.d.
Potassium	177	173	161	186
Protein solubility, %	81	61	78	48
TIA g/kg	3.3	<0.5	5.1	<0.5

Abbreviations: TIA, trypsin inhibitor activity.

**Table 3 animals-11-02668-t003:** Growth performance characteristics of the broiler chickens.

Variety	Heat-Stable		Heat-Labile			*p*-Value		
Heat Treatment	Low	High	Low	High	SEM	Heat	Variety	H × V
Animals at start, n	84	84	84	84				
Animals at day 15, n	80	79	81	77				
Animals at day 36, n	75	71	79	72				
Ratio male/female at day 36, %	51/49	48/52	39/61	54/46				
Initial body weight, g	39	40	40	40	0.17	0.884	0.810	0.787
Starter period, day 1–14								
BW on day 14, g	403 ^b^	405 ^b^	381 ^a^	395 ^ab^	2.88	0.119	0.003	0.194
ADG, g/day	28 ^b^	28 ^b^	26 ^a^	27 ^ab^	0.22	0.128	0.003	0.195
ADFI, g/day	35	35	33	34	0.23	0.396	0.011	0.596
FCR, kg/kg	1.26	1.27	1.28	1.28	0.01	0.791	0.390	0.890
Grower period, day 15–28								
BW on day 28, g	1330 ^b^	1306 ^b^	1214 ^a^	1290 ^ab^	13.56	0.243	0.006	0.032
ADG, g/day	66 ^b^	64 ^ab^	59 ^a^	64 ^ab^	0.83	0.355	0.018	0.038
ADFI, g/day	88 ^b^	87 ^b^	80 ^a^	87 ^b^	0.90	0.058	0.017	0.010
FCR, kg/kg	1.35	1.37	1.36	1.37	0.01	0.464	0.786	0.805
Finisher period, day 29–36								
BW on day 36, g	2044 ^b^	2080 ^b^	1860 ^a^	2055 ^b^	24.45	0.004	0.007	0.035
ADG, g/day	89 ^ab^	97 ^b^	81 ^a^	96 ^b^	2.17	0.007	0.205	0.333
ADFI, g/day	147 ^b^	147 ^b^	134 ^a^	150 ^b^	1.96	0.016	0.128	0.024
FCR, kg/kg	1.65	1.57	1.66	1.63	0.02	0.152	0.382	0.555
Total, day 1–36								
ADG, g/day	57 ^b^	58 ^b^	52 ^a^	58 ^b^	0.70	0.004	0.007	0.034
ADFI, g/day	81 ^b^	81 ^b^	75 ^a^	82 ^b^	0.79	0.014	0.018	0.008
FCR, kg/kg	1.45	1.43	1.46	1.46	0.01	0.563	0.273	0.573
Mortalities, %	3.6	9.4	4.6	9.5	1.3	0.049	0.816	0.864

Abbreviations: BW, body weight; ADG, average daily gain; ADFI, average daily feed intake; FCR, feed conversion rate; superscript letters indicate significant differences between treatments (*p* < 0.05)

**Table 4 animals-11-02668-t004:** Slaughter characteristics of the broiler chickens.

Variety		Heat-Stable		Heat-Labile				*p*-Value		
Heat Treatment	n	Low	High	Low	High	SEM	Heat	Variety	Sex	H × V
Body weight (BW), g	297	2055 ^b^	2096 ^b^	1884 ^a^	2053 ^b^	15.85	0.004	<0.001	<0.001	0.040
Eviscerated carcass (EC), g	297	1601 ^b^	1639 ^b^	1449 ^a^	1595 ^b^	0.13	0.002	<0.001	<0.001	0.043
Abdominal fat, g/100 g EC	297	0.74 ^b^	0.75 ^b^	0.52 ^a^	0.65 ^ab^	0.02	0.127	<0.001	<0.001	0.167
Heart, g/100 g EC	297	0.71 ^ab^	0.65 ^a^	0.76 ^b^	0.72 ^b^	0.01	0.014	<0.001	<0.001	0.472
Liver, g/100 g EC	297	2.58 ^a^	2.55 ^a^	2.83 ^b^	2.73 ^ab^	0.03	0.272	<0.001	0.146	0.615
Gizzard, g/100 g EC	297	1.55	1.34	1.36	1.32	0.05	0.203	0.287	0.807	0.408
Pancreas, g/kg EC	142	2.68 ^bc^	2.34 ^a^	2.93 ^c^	2.58 ^ab^	0.04	<0.001	<0.001	<0.001	0.775
Chilled Carcass (CC), g	274	1601 ^b^	1622 ^b^	1472 ^a^	1585 ^b^	12.55	0.015	<0.001	<0.001	0.067
Dressing, g CC/100 g BW	274	77	77	77	77	0.11	0.915	0.062	0.661	0.928
Carcass for grilling, g	274	1438 ^b^	1472 ^b^	1334 ^a^	1437 ^b^	12.02	0.012	<0.001	<0.001	0.173
Breast, g/100 g CC	48	29.3 ^b^	27.0 ^ab^	28.2 ^ab^	26.0 ^a^	0.36	0.001	0.151	0.267	0.751
Thigh and drumstick, g/100 g CC	48	24.3	25.5	24.9	23.2	0.21	0.822	0.475	0.710	0.221
Wings, g/100 g CC	48	9.6	10.1	9.6	10.7	0.20	0.043	0.422	0.444	0.405
Remainder of carcass, g/100 g CC	48	27.4	28.1	27.9	28.2	0.19	0.207	0.464	0.250	0.439

Abbreviations: BW, body weight; EC, eviscerated carcass; CC, chilled carcass; superscript letters indicate significant differences between treatments (*p* < 0.05)

**Table 5 animals-11-02668-t005:** Effect of the diets applied on apparent total tract digestibility.

Variety	Heat-Stable		Heat-Labile			*p*-Value		
Heat Treatment	Low	High	Low	High	SEM	Heat	Variety	H × V
Dry matter	0.959	0.957	0.954	0.958	0.001	0.499	0.341	0.135
Ash	0.896	0.897	0.893	0.897	0.002	0.460	0.634	0.613
N retention	0.704 ^a^	0.646 ^b^	0.672 ^ab^	0.666 ^ab^	0.007	0.025	0.635	0.061
Ether extract	0.856	0.856	0.850	0.861	0.003	0.294	0.890	0.267
Gross energy	0.750	0.736	0.723	0.739	0.004	0.915	0.143	0.059
Starch	0.948 ^a^	0.940 ^b^	0.941 ^b^	0.937 ^b^	0.001	0.004	0.008	0.271

^a,b^ indicate significant differences between treatments (*p* < 0.05).

**Table 6 animals-11-02668-t006:** Effect of diets applied on pre-caecal apparent amino acid digestibility.

Variety	Heat-Stable		Heat-Labile		SEM	*p*-Value		
Heat Treatment	Low	High	Low	High		Heat	Variety	H × V
Dry matter	0.712	0.691	0.693	0.681	0.011	0.162	0.202	0.694
Crude protein	0.768	0.724	0.752	0.746	0.013	0.065	0.801	0.151
Met	0.892	0.872	0.888	0.892	0.008	0.334	0.311	0.139
Cys	0.705	0.646	0.681	0.658	0.016	0.018	0.733	0.281
Lys	0.817	0.781	0.812	0.800	0.013	0.071	0.574	0.343
Thr	0.741	0.710	0.723	0.729	0.014	0.252	0.778	0.273
Arg	0.853	0.835	0.846	0.851	0.010	0.530	0.680	0.271
Ile	0.783	0.766	0.774	0.783	0.012	0.734	0.732	0.301
Leu	0.783	0.763	0.771	0.781	0.012	0.656	0.803	0.224
Val	0.775	0.754	0.764	0.770	0.013	0.566	0.872	0.288
His	0.810	0.776	0.802	0.792	0.011	0.066	0.745	0.276
Phe	0.811	0.792	0.802	0.811	0.011	0.651	0.663	0.245
Gly	0.729	0.688	0.713	0.703	0.015	0.100	0.994	0.302
Ser	0.771	0.739	0.753	0.753	0.013	0.217	0.881	0.228
Pro	0.805	0.775	0.796	0.784	0.010	0.052	0.985	0.372
Ala	0.775	0.741	0.759	0.761	0.013	0.222	0.880	0.160
Asp	0.784 ^a^	0.728 ^c^	0.775 ^ab^	0.732 ^bc^	0.012	0.000	0.805	0.566
Glu	0.844	0.817	0.836	0.827	0.009	0.060	0.898	0.328
NH_3_	0.759	0.739	0.752	0.743	0.012	0.228	0.897	0.640
Sum ^x^	0.800	0.770	0.790	0.783	0.011	0.117	0.902	0.301

^x^ Sum of amino acids (without NH_3_); ^a,b,c^ indicate significant differences between treatments (*p* < 0.05).

## Data Availability

Not applicable.

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
