# Peer review of "Interaction of Soybean Varieties and Heat Treatments and Its Effect on Growth Performance and Nutrient Digestibility in Broiler Chickens"

_animals, 2021, doi:10.3390/ani11092668_

Round 1

Reviewer 1 Report

This is a very nice and well-written paper, with clear objectives, good discussion and correctly prepared results. The topic is interesting and the paper is worth of publication.

I have a few questions for clarification and some suggestions.

Title:

-This is up to you of course, but I suggest modifying the title to for example: Interaction between A and B, A interacts with B or the effect of A depends on B, because your central finding is about the interesting interaction between the two factors where different soybean varieties respond differently to heat treatment.

Simple summary:

-L-14:  interfering negative with: correct to interfering negatively with…

Abstract:

-L25-26: Only the effect of different soybean varieties was investigated? What about heat treatment?

-Please provide information about feed form, i.e. mash? steam pelleted?

-Please provide info about the length of the experiment

-Please provide info about replication/treatment

Introduction:

-Please provide examples of the common heat treatment method of the soybean with temperature ranges…

Material and methods:

-L81: what is the purpose of this sentence "As usual under practical conditions..."

-L82: how many replicates?

Performance Parameters

-L114: birds in each pen/ birds from each pen

-L115, there should a comma after slaughtering..

Sample Collection and Chemical Analysis

-L127: how did you collect excreta when you had wood shaving as litter? why only one day collection and not more? e.g. three days?

-L143: The reference for the GFE 1999 in the reference list is missing.

Table 1:

-How many Units of phytase in a kg diet?

-Dicalcium phosphate, under high stable low/heat: remove the 0 next to 09.4

 Table 2:

-Is this trypsin inhibitors concentration (3.3 g/kg) or trypsin inhibitor activity 3.3 g/kg. What is the difference between the two and how can you measure enzyme activity in g/kg?

Table 5:

-P value (interaction) for N retention was 0.0613 and there are letters (a, b) to separate means.

-P value (interaction) for Gross energy was 0.0592 but no letters to separate the means. Please add letters.

Discussion:

-L249: what do you mean by indicating growth depression? do you mean causing growth depression?

-L261-264: “Lower nitrogen retention for high-temperature-treated soybean cakes indicates 261 lower protein digestibility for these treatments, although no negative effect on growth 262 performance was observed. These findings might confirm sufficient supply of essential 263 amino acids in all experimental diets.” 

Does this mean that the diets were formulated to contain excessive amounts of amino acids (i.e. more than the requirements)? If so, this would confound the results. Please discuss this point.

-L273: "One reason for the observed differences in starch digestibility may be the fact that amylose–lipid complexes were generated in the high-temperature-treated varieties [28] " In this referenced paper, heat treatment was referred to as pelleting, extrusion and expansion, and all these feed treatment are considered a very short heat treatment methods unlike the one used in the paper 110 -120C for 20 min and more and under pressure.

It may be plausible that maillard reaction took place in the high temp. treated varieties (although both treatment are considered high for the occurance of maillard reaction) but the difference in diet starch digestibility, although significant, did not exceed (on average) the 0.64%. So if maillard reaction was a contributing factor, I reckon differences in starch digestibility should be higher and not that low.

Author Response

Dear Editor in Chief,

First, we want to thank you and the reviewers for the comments and useful advice to improve our manuscript. We have taken into account all the suggestions and proposed improvements in the revised manuscript.

Reviewer 1

Title:

-This is up to you of course, but I suggest modifying the title to for example: Interaction between A and B, A interacts with B or the effect of A depends on B, because your central finding is about the interesting interaction between the two factors where different soybean varieties respond differently to heat treatment.
Thanks for the input – we adapted the title

Simple summary:

-L-14:  interfering negative with: correct to interfering negatively with…
done 

Abstract:

-L25-26: Only the effect of different soybean varieties was investigated? What about heat treatment?
Thanks for the hint. We corrected this in the updated manuscript (p.1/L.27)

-Please provide information about feed form, i.e. mash? steam pelleted?
Diet form was mash. We added the information in the abstract (p.1/L.30)

-Please provide info about the length of the experiment
done (p.1/L.30)

-Please provide info about replication/treatment
done (p.1/L.31)

Introduction:

-Please provide examples of the common heat treatment method of the soybean with temperature ranges…

Samples of common heat treatment methods were included in the modified version of the manuscript (p.2/L.69-73)

Material and methods:

-L81: what is the purpose of this sentence "As usual under practical conditions..."
We wanted to point out that we used both sexes in our experiment, which is common in poultry production however not always the case in scientific studies, where sexed animals were used). A further element of this study was an analysis of the gut microbiota (the manuscript is under preparation), which was an additional reason to use both, males and females.

-L82: how many replicates?
6 replicates. Information is now added in the manuscript. (p.3,L.96)

Performance Parameters

-L114: birds in each pen/ birds from each pen
Thanks, is now corrected.

-L115, there should a comma after slaughtering..
done

Sample Collection and Chemical Analysis

-L127: how did you collect excreta when you had wood shaving as litter? why only one day collection and not more? e.g. three days?
Litter was removed from the pens for purpose of excreta collection. Unfortunately, only one collection day was possible at the research station for technical reasons.

-L143: The reference for the GFE 1999 in the reference list is missing.
Reference was added in the new manuscript.

Table 1:

-How many Units of phytase in a kg diet?
We used the Optiphosâ 2500 CT 6-phytase derived from E. coli (Huvepharma) in a dosage of 0.1 g per kg diet. Resulting in 250 OTU per kg diet. We included this information in the revised manuscript. (p.5/L.183)

-Dicalcium phosphate, under high stable low/heat: remove the 0 next to 09.4
done

 Table 2:

-Is this trypsin inhibitors concentration (3.3 g/kg) or trypsin inhibitor activity 3.3 g/kg. What is the difference between the two and how can you measure enzyme activity in g/kg?
Trypsin inhibitor activity is the standard method to measure trypsin inhibitors. It expresses the amount of trypsin inhibitor, which is inhibited by the diet/feedstuff.

Table 5:

-P value (interaction) for N retention was 0.0613 and there are letters (a, b) to separate means.

-P value (interaction) for Gross energy was 0.0592 but no letters to separate the means. Please add letters.
We placed the letters following the tukey Kramer test between the respective treatments. Differences between treatments were significant for N retention and starch digestibility, but not for gross energy.

 Discussion:

-L249: what do you mean by indicating growth depression? do you mean causing growth depression?
Thanks for the helpful comment. The wording of this sentence was flawed and corrected. (p.11/L.281)

-L261-264: “Lower nitrogen retention for high-temperature-treated soybean cakes indicates 261 lower protein digestibility for these treatments, although no negative effect on growth 262 performance was observed. These findings might confirm sufficient supply of essential 263 amino acids in all experimental diets.” 

Does this mean that the diets were formulated to contain excessive amounts of amino acids (i.e. more than the requirements)? If so, this would confound the results. Please discuss this point.
Diets were calculated to meet the recommendation of AVIGAN for Ross 308 broilers (2014). Hence a safety margin for the calculated nutrients can’t be excluded. Hence ´Confirm´ is not the proper word for this sentence – thanks for the hint. We rephrased this sentence in the new manuscript. (p.10/L.299)

-L273: "One reason for the observed differences in starch digestibility may be the fact that amylose–lipid complexes were generated in the high-temperature-treated varieties [28] " In this referenced paper, heat treatment was referred to as pelleting, extrusion and expansion, and all these feed treatments are considered a very short heat treatment methods unlike the one used in the paper 110 -120C for 20 min and more and under pressure.
We added new references, which should be more suitable. (p.10/L.312)

It may be plausible that maillard reaction took place in the high temp. treated varieties (although both treatment are considered high for the occurance of maillard reaction) but the difference in diet starch digestibility, although significant, did not exceed (on average) the 0.64%. So if maillard reaction was a contributing factor, I reckon differences in starch digestibility should be higher and not that low.
We added a note, pointing out that differences for starch digestibility were numerically low in the new manuscript.(p.10/L.310)

Reviewer 2 Report

Dear Authors,

my comments are below:

check editorial errors (double spaces, gaps in the text, etc.)
line 35: P <0.10)? Should it be 0.10? Is that a bug?
keywords: soybean instead of Soybean
Literature or literature?
in the Introduction section, line 62: TIA - enter the entire name for the first time in this section.
The introduction is quite short. The authors should provide more information, e.g. on previous activities in terms of the digestibility of individual nutrients, referring to what was done in the presented research.
why were both sexes used, no division? Please add it in the text (line 81).
why were the birds kept 36 days and not 42?
describe the exact slaughter of the birds.
describe in the methodology how ADG, ADFI, FCR etc. were calculated
add a subsection statistical analysis.
what post-hoc test was used.
in the results, the tables should be placed directly below the text that relates to the individual results.
table 1. why there is no data in the Lys-Arg lines?

describe in the tables the full names of the components (P, Ca, Na, K ... etc).
numbers in tables should be consistently described (decimal places. 81.00 1.45 3.60 ... or otherwise, at your discretion.
why SEM is described as "0.00087426"

the discussion is quite "poor". The topic is quite known and taken up in many scientific activities and deserves a larger description, indicating the mechanisms of action of individual dependencies and features. The work is interesting but requires correction.
Supplementary Materials:? empty?
Literature is not properly formatted.

The work requires many editorial corrections and substantive refinement of the text.

Yours sincerely,

Reviewer.

Author Response

Dear Editor in Chief,

First, we want to thank you and the reviewers for the comments and useful advice to improve our manuscript. We have taken into account all the suggestions and proposed improvements in the revised manuscript.

Reviewer 2:

check editorial errors (double spaces, gaps in the text, etc.)
Thanks for the comment, we have searched for editorial errors, they are now corrected in the new manuscript.
line 35: P <0.10)? Should it be 0.10? Is that a bug?
No, actually ´p<0.10´ is what we have applied.
keywords: soybean instead of Soybean
Thanks for the comment. We corrected the word in the revised manuscript. (p.1/L.41)
Literature or literature?
done. (p.2/L.62)
in the Introduction section, line 62: TIA - enter the entire name for the first time in this section.
Thanks for the note. We corrected this error. (p.2/L.65)
The introduction is quite short. The authors should provide more information, e.g. on previous activities in terms of the digestibility of individual nutrients, referring to what was done in the presented research.

In the modified version of the manuscript, we tried to include this recommendation. For example, please see page 2, line 63 and following.

why were both sexes used, no division? Please add it in the text (line 81).
We used both sexes because this is the way birds are fattened in practice. A further element of this study was an analysis of the gut microbiota (the manuscript including corresponding results is under preparation), which was an additional reason to use males and females.
why were the birds kept 36 days and not 42?
To our knowledge, this is the standard fattening period in Austria and in other countries in Europe. Therefore, we also chose this period length. Nevertheless, the effects of the treatments were sufficiently pronounced, independent of a 35- or 42-day fattening period.
describe the exact slaughter of the birds.
Information for stunning and killing is now provided in the manuscript (p.3/L.103)
describe in the methodology how ADG, ADFI, FCR etc. were calculated
done (p.3/L.130)
add a subsection statistical analysis.
done (p.4/L.169)
what post-hoc test was used.
The Tukey Kramer test was applied. We added this information in the revised manuscript. (p.4/L.176)
in the results, the tables should be placed directly below the text that relates to the individual results.
done
table 1. why there is no data in the Lys-Arg lines?
Thanks, we added the values in the table.
describe in the tables the full names of the components (P, Ca, Na, K ... etc).
done
numbers in tables should be consistently described (decimal places. 81.00 1.45 3.60 ... or otherwise, at your discretion.
done
why SEM is described as "0.00087426"
Indeed, we are sorry for using too many digits, thanks for the comment. The new manuscript only includes 3 decimal places.

the discussion is quite "poor". The topic is quite known and taken up in many scientific activities and deserves a larger description, indicating the mechanisms of action of individual dependencies and features. The work is interesting but requires correction.

Thank you for this frank comment. The discussion section in the revised manuscript has been rewritten and enlarged accordingly. We hope that the new version is now improved ín a way to make this paper more attractive to potential readers.

Supplementary Materials:? empty?
The section ´supplementary materials´ was not supposed to be listed in the manuscript. It has been removed in the revised manuscript.
Literature is not properly formatted.
Literature section has been updated now.

Reviewer 3 Report

The trial is interesting because soybeans from Europe are used. Why were small birds culled on day 15? I was also wondering, has the equal number of sexes in the pen some effects on the results.

Introduction:

I would like to see more results of different processing condition and their effects on nutrition.

Material and methods:

Write the temperature and lighting regimes more clearly.

Why were the light birds removed from the trial on day 15?

How were the soybeans de-oiled?

I do not understand what means “in an autoclave at 110C for 20 min plus 20 min for heating and cooling”

How was the ileal digestibility calculated?

Table 1. Was the diet amino acids analysed?

Discussion:

I would like to see more studies with different heat treatments

Author Response

Dear Editor in Chief,

First, we want to thank you and the reviewers for the comments and useful advice to improve our manuscript. We have taken into account all the suggestions and proposed improvements in the revised manuscript.

Reviewer 3:

The trial is interesting because soybeans from Europe are used. Why were small birds culled on day 15? I was also wondering, has the equal number of sexes in the pen some effects on the results.
Regarding the limited number of animals per pen, we wanted to ensure homogenous performance among the animals. One should be aware of the fact, that only one or two outliers can influence the results considerably at such a low number of animals per pen.

Introduction:

I would like to see more results of different processing condition and their effects on nutrition.

The introduction of the revised manuscript now includes more information about different processing conditions and their effects on nutrition. For example, see page 3, lines 63-72

Material and methods:

Write the temperature and lighting regimes more clearly.
done

Why were the light birds removed from the trial on day 15?
See above

How were the soybeans de-oiled?
We used a continuous screw press.

I do not understand what means “in an autoclave at 110C for 20 min plus 20 min for heating and cooling”
Temperature was held at 110°C for 20 minutes. But also contributing to the heat treating effect were the 20 Minutes needed for heating up the autoclave with soybeans to 110°C and 20 minutes needed to cool it down before autoclave could be reopened.

How was the ileal digestibility calculated?
Thanks for the comment. Respective formula was added to the new manuscript (in combination with the formula for the total tract digestibility). (p.4/L.167)

Table 1. Was the diet amino acids analysed?
Diet was calculated with tabular values available for soybean cakes. Finisher diets were also analysed for amino acid content afterwards, to calculate the digestibility of amino acids.

Discussion:

I would like to see more studies with different heat treatments

The discussion in the new manuscript has been rewritten and improved according to the reviewers’ recommendations. (p.10/L.284 and p.10/L.305)

Round 2

Reviewer 3 Report

I think the manuscipt is ready for publication.